Analysis

# The billion-dollar case for sustaining palaeontology's digital databases

**Elizabeth M. Dowding** [1] ✉, **Emma M. Dunne** [1,2] ✉, **Katie S. Collins** [3], **Katheryn Cryer** [1], **Kenneth De Baets** [4], **Danijela Dimitrijević** [1], **Stewart M. Edie** [5], **Seth Finnegan** [6,7], **Wolfgang Kiessling** [1], **Kari Lintulaakso** [8], **Lee Hsiang Liow** [9,10], **Holly Little** [5], **Lin Na** [11], **Shanan E. Peters** [12], **Johan Renaudie** [13], **Erin E. Saupe** [14], **Barbara Seuss** [1], **Jocelyn A. Sessa** [15], **Jansen A. Smith** [16], **Mark D. Uhen** [17], **John W. Williams** [18] **& Ádám T. Kocsis** [1] ✉

The digital revolution has transformed palaeontology through the development of openly accessible, community-driven databases that underpin some of the most complex and large-scale empirical studies of the history of life on Earth. These systems safeguard high-effort, volunteered data and have revealed major macroevolutionary patterns, including the 'Big 5' mass extinctions. These efforts also represent remarkable global scientific and financial investment, which is continually required to support the next generation of databases and associated research. Here we conducted a survey of 118 palaeontological and allied Earth science databases, analysing their diversity dynamics, including origination and extinction rates. We show that approximately 85% of all community-curated databases have lifespans of less than 15 years, putting decades of investment at risk. We show that database creation effort has increased in the past 30 years, with peaks in database loss related to 5-year funding cycles. We advocate for strategies to enhance database longevity, including sustained funding models, stronger institutional support and modular backend architectures that better link international community databases to each other and to fossil specimens.

The study of the history of life on Earth is inherently multidisciplinary and conducted at various scales from local to global. This scientific inquiry draws from geology, biology, chemistry, archaeology and mathematics, among others, to breach human temporal perspectives, reconstruct ancient ecosystems, investigate the drivers of biodiversity and forecast how life will respond to today's changing environments[1–3]. The fossil record is essential for understanding biodiversity and Earth system processes operating at timescales beyond the twentieth and twenty-first century window of instrumental observations. The past also provides examples of Earth system states with instructive analogies to the societally novel climates that are now emerging[4,5]. From

their very beginning, palaeontological databases (see 'Glossary' in Supplementary Table 1) have played pivotal roles in enabling the field to scale up from site-level studies to global-scale research. These databases were founded by scientists seeking to address questions beyond the scope of any individual palaeontological dataset, including identifying global mass extinctions and their roles in macroevolution[6] and the earliest evidence of climate-driven species' range shifts and ecosystem transformations[7,8]. The subsequent migration of palaeontological databases to open-access online platforms and data systems (encompassing the database, its system for community governance and data curation, and any associated software services) increased their accessibility

**Fig. 1 | Palaeontological information in an Earth system context.** From left to right, planetary or global-level information can be used to understand tectonic processes, climate and landscape evolution, and eco-evolutionary processes across timescales ranging from the present to billions of years. Outcrop- or borehole-level data provide local- to regional-scale time series that can be used to reconstruct climate, geochronology (age), sea level fluctuations and community dynamics. Finally, specimen-level data are the foundational unit in palaeobiology for analyses of, for example, taxonomy, biotic interactions, geochemistry, functional ecology, behaviour and taphonomic processes. Credit: Science Graphic Design.

and amplified their impact by enabling new questions to be explored, broader collaboration and reproducibility[9,10].

Today, openly accessible, community-run data systems function as collective archives for scientific data and knowledge about the history of life on Earth[11] (Fig. 1). These databases are invaluable for quantitatively reconstructing ancient ecosystems[12], tracing evolutionary pathways[13], studying climate- and human-driven eco-evolutionary dynamics at continental to global scales[5,14–16], and predicting future biological and geological changes—or at least assessing the limits to predictability in an increasingly novel world[17,18]. By integrating these palaeontological databases with other open data systems, scientists can tackle increasingly complex, multifaceted questions that are top priorities in global change research[3,19,20].

Representing developers, leaders, curators and users of 15 community-run palaeontological databases (Supplementary Table 2), we review the current landscape of palaeontological data systems to assess the volume, variety and value of data held in these community-curated, openly accessible databases, their diversity dynamics and longevity, the challenges faced and the opportunities for sustainable growth and scientific discovery. We close by providing recommendations for continued investment from researchers, maintainers, developers and funders.

## An overview and history of palaeontological data and databases
### Key concepts
Palaeontology aims to reconstruct the history of life across the broadest possible range of spatiotemporal scales and throughout the geological record (Fig. 1). Here, palaeontology encompasses closely related fields, including but not limited to palaeobiology, biostratigraphy and palaeoecology. As our collective understanding of geological processes evolves, new scientific questions emerge, and our interpretation of the fossil record is updated. This, in turn, affects our understanding of the processes that we infer from it and drives new primary-data collection campaigns (for example, fieldwork) and the reinterpretation and reanalysis of existing data. Examples include taxonomic reidentification of old fossils following new finds[21], redating of core samples and refinement of the geological timescale using newer and improved methods and data (for example, ref. 22), re-interpretation of environmental/

depositional contexts (for example, ref. 23) and incorporation of palaeobiogeographic patterns into tectonic models (for example, ref. 24).

Palaeontologists work with two primary forms of data: 'fundamental data' and 'processed data'. Fundamental data are direct observations and sampling of the sedimentary record and fossil specimens within these sediments. Examples of fundamental data include geospatial locations, physical samples, multimedia recording, counts and geochemical analysis. When these fundamental data are subject to further interpretation, such as through taxonomic study and analyses of morphology, preservation and biotic associations, they are translated into processed data. For example, within database structures, age controls (for example, radiocarbon dates) are fundamental data, and age-depth models (used to estimate the age of different depths within a sediment core or stratigraphic profile) are processed data and are frequently revised. Although fundamental and processed data exist on a continuum, whenever possible, palaeontological databases should maintain the strongest links to fundamental data and the associated physical samples or specimens. A focus on fundamental data and well-established provenancing is essential for reproducibility and resampling efforts (for example, when palaeontologists remeasure a fossil or reassess its taxonomic identity). A focus on fundamental data also reduces database maintenance costs, because of the frequent revisions associated with processed data. Lastly, good provenancing can ensure against corollary risks such as 'data cannibalism'[25,26] when databases are used as data sources for other, secondary databases without proper attribution and dataset-level provenancing, which can violate the standard CC-BY licences that accompany most open-access data resources.

### Database development history
**The past—first-generation research databases.** First-generation compilations of palaeontological data usually were launched with a specific research question or other purpose and focussed on the collation of processed data. For example, the collation inferred temporal (that is, stratigraphic) distribution of fossil taxa using harmonized taxonomic lists across sites, which are the minimum requirement for assessing the history of biodiversity (for example, refs. 27,28) and the shifting distribution of taxa across space and time[8,29]. These were initially collated as physical repositories (for example, the John Williams

Index of Palaeopalynology[30]) or as offline digital entities (for example, Sepkoski's Compendium[28,31] and the first version of the Neptune Sandbox database (NSB)[30]). These first-generation databases were often built either by individual scientists over their careers or by small research teams.

**The present—second-generation multipurpose and community data systems.** As the field advanced, palaeontologists gained further understanding of the various factors that distort the structure of the fossil record (for example, ref. 32), new research questions emerged (for example, reconstruction of past biomes and terrestrial carbon sequestration[33]), and palaeontologists developed new quantitative methods to address emergent questions. These, in turn, led to new efforts to reanalyse existing databases. For example, in deep-time biodiversity studies, the field progressed from recording observed first- and last-appearance dates of taxa[31] to the recording of fossil occurrences from the entire stratigraphic record (for example, refs. 13,34). As the breadth of questions increased, second-generation data systems (for example, Paleobiology Database (PBDB)[34] and Neotoma[10]) also began to store an increasing variety of fundamental data types, including the geographic coordinates of fossil sites, taxon abundance and traits, stratigraphic position, lithological characteristics and environmental covariates.

In parallel to this expansion of database scope, the leadership and development of these databases increasingly shifted from a few individual experts to community-curated data systems. For example, in Quaternary palynology, individual efforts to build databases and map continental-scale plant distributions for North America and Europe[8,29] expanded to continental-scale databases around the world, each with their own data leaders and stewards[10,35–37].

The data structures of current, second-generation databases vary substatially, reflecting their founding aims and user communities. As examples, PBDB was originally developed to enable, among other things, sampling-standardized estimates of Phanerozoic diversity[13]; NOW (New and Old Worlds database of fossil mammals) focused on Cenozoic mammal macroevolution[38]; Neotoma was designed to study species range shifts during the Quaternary glacial–interglacial cycles across multiple taxonomic groups[10,36–38]; and the Geobiodiversity Database (GBDB) was designed to support high-resolution stratigraphic data by linking fossil occurrences to detailed geological sections[39].

All these databases continue to grow in scope and incorporate new kinds of data. As new questions emerge and data continue to diversify and increase in accessibility[25,40], the range of scientific applications of these second-generation databases far surpass their original scope and yield input for thousands of scientific studies (see 'Database use' in Supplementary Data). For example, PBDB occurrence data have been used for climatic modelling[41,42], landscape evolution[43] and palaeogeographic models[24]. Similarly, NOW data have been used to study macroevolutionary expansion[44] and Neotoma data for reconstructing past climates[45], constraining past land cover dynamics and the terrestrial carbon cycle[46], and documenting cross-continental species invasions[47]. The scientific utility and applications of these databases thus continue to grow and diversify, as do the databases themselves.

**The near future—from databases to third-generation data systems**

Palaeontology is poised for its next transformative phase, in which second-generation databases will better integrate with each other and with other components of the palaeontological, Earth and life sciences data infrastructures, to address more integrative, cross-disciplinary and multiscalar questions. The transition to the third-generation database systems has already begun, with cross-database integration a core focus of backend development, using, for example, the linking capabilities provided by new data structures such as LinkML (https://linkml.io/). Other efforts are focusing on improved efficiency and more

sustainable codebases through modular design[48–50]. The development of integrative platforms, such as Deep-time Digital Earth[51], and the continued growth of existing databases to support new data types, such as ancient environmental DNA[52], are striking movements towards third-generation databases.

These efforts towards integration and efficiency will enable new scientific questions to be answered at increasing power. For example, in the Big Questions in Paleontology Project[53], representative questions include 'How do external environmental drivers (for example, plate tectonics, global temperature and sea level) influence the structure of biological systems at different spatiotemporal scales?', 'How does the prevailing climate state experienced by species and communities influence their response to perturbation?' and 'To what extent are the phases of events (for example, collapse and recovery) during extinctions consistent across different biotic crises?' Addressing these integrative questions requires scalable, connected data that capture, for example, phenotypic variation among individuals in a population, in conjunction with high-stratigraphic-resolution, palaeoenvironmental and specimen-level information. These scientific needs demand further advances in how palaeontological data are reported, structured, integrated, managed and sustained. Cross-institutional aggregation of museum specimen information into iDigBio[54] and the Global Biodiversity Information Facility (GBIF)[55] provide models of how biodiversity databases can grow and be enhanced by ever-improving biodiversity data standards, such as the Darwin Core[56] and ABCDEFG[57], featuring a stronger focus on available metadata[58].

With the growth of these databases has come an awareness of their importance and impact beyond simply answering scientific research questions. Careers of an entire generation of scientists are now influenced by publicly accessible, interoperable data, and access to international, high-quality data has led to the rise of quantitative subdisciplines in palaeobiology[59,60]. Similarly, in allied fields such as geochemistry, the advent of open scientific databases PetDB and GEOROC has enabled the rise of statistical geochemistry[61]. At the same time, new concerns have arisen about whether these databases encode and perpetuate past and present inequities, such as parachute science[62], and how best to reduce these inequities to truly fulfil the deeper mission of these databases to ensure democratized data access for all[61–65]. To address these issues, the concept of community governance of palaeontological databases must be further broadened to include additional voices and to develop more effective, context-sensitive strategies that address issues of access, reciprocal research and data equity[51,53,66–68].

## Landscape survey of the current state and valuation of palaeodata

### Survey and assessment of palaeodata

An online survey of available palaeontological and Earth science databases was conducted using search terms in multiple languages to identify 'community-run' databases (Supplementary Table 4). Community-run databases were required not to be affiliated with any state governing body, including state-funded museums or geological surveys, and are considered 'open access' by virtue of making their data freely available for general use. The period of activity was identified by the first publication of the database in the peer-reviewed scientific literature, and its endpoint was identified through the last update to the web service, data repository and/or latest published article. Among the 171 palaeontological and Earth systems databases identified, 118 were open access and community-run, and their extinction rate, origination rate and diversity dynamics were assessed (Fig. 2).

We assessed the replacement value of the data stored in three databases: PBDB, GBDB and Neotoma. We followed Thomer et al.[69] and estimated conservatively a value of US$3,000 per collection (Methods). This valuation is clearly an underestimate, not only because it does not cover all costs, but also because it assumes that the sample localities are still accessible and have not been destroyed by human

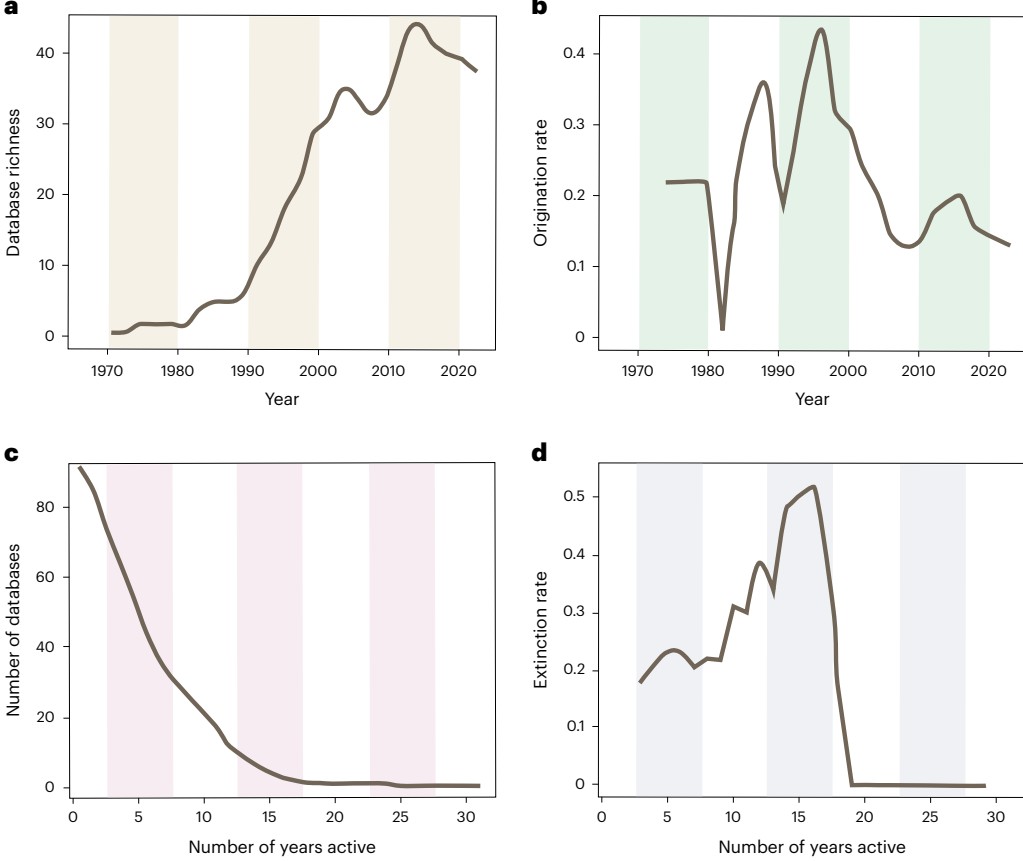

**Fig. 2 | Diversity dynamics of 118 community-developed palaeontological databases from the 1970s to 2024. a**, The range-through richness of databases by year. **b**, The origination rate of databases through time, indicating areas of peak activity for novel database development between 1995 and 2005. **c**, Diversity of databases as a function of years active (that is, database survivorship) showing the loss of >80% of database diversity by 10 years of activity. **d**, The rolling mean per-capita extinction rate of databases as a function of years active since inception, with peaks at 5, 15 and 25 years of activity.

land use or natural processes such as earthquakes. The data from inaccessible locations are therefore irreplaceable and priceless[70–72]. Research effort, storage, maintenance, curation and expertise were not calculated, resulting in conservative values that do not cover the entire cost to replace the extant data. They additionally do not cover the costs of labour, server hosting and infrastructure development that go into setting up and sustaining databases. Furthermore, results do not cover the article processing costs to publish a scientific paper (mean US$2,300)[73] or the value of papers published (estimated at over US$5,000 per item)[74].

### Historical trends in results

Based on our web search, database origination rates peaked in the 1970s and 1990s, with a tertiary peak in the 2010s (Fig. 2). Nearly 50% of databases (*n* = 118) became inactive within just 5 years, and fewer than 15% survived a full decade. Only a rare 5% remained active for over 15 years (Fig. 2). This 5-year interval coincides with the standard competitive funding cycles of many large research grants from wealthy international unions or countries with a high gross domestic product, such as through the European Research Council or the US National Science Foundation, respectively. This means that, after 5 years, up to 65% of value-added data effort—representing years of data aggregation, data harmonization and cleaning, technical development and scientific labour—is left unmaintained and sometimes inaccessible.

For example, recent attacks on the cyber infrastructure of the Museum für Naturkunde Berlin resulted in the loss of community access to the NSB. The NSB was intermittently funded (16 funded years since 1990) and maintained by an individual expert (see 'Curator

review' in Supplementary Data). The NSB held hundreds of thousands of marine plankton microfossil species that are used to research marine community responses to climate change and has been a data contributor to other databases, including BioDeepTime[75], Microtax[76], GBDB[39] and Triton[77] among others. This attack not only impacted a key resource for microfossil taxonomists, evolutionary (palaeo)biologists and palaeoceanographers, but the data provenance of the dependent databases was compromised. The lack of funding and dedicated technical support resulted in insufficient failsafes at the museum. Instead, through community activity, external versions of the NSB—for example, those hosted on Zenodo[78]—are contributing to database recovery, further highlighting the value of community contributions in sustaining data resources.

Although some database development efforts are intended for short-term use and do not assume database longevity, the loss of these databases is not just a scientific concern; it also represents a substantial economic waste (Fig. 2 and Table 1). The cost of allowing valuable data infrastructure to degrade is not conceptual but quantifiable and substantial. The best-case scenario for at-risk databases, as illustrated by strategy 2, involves integration with larger data systems—an example being the current incorporation of 34 constituent databases into Neotoma. The data protected and expanded by Neotoma were recently estimated to cost over US$1.5 billion to replace[69]. Despite the valuation and proven utility to the community[59,68,79], even long-lasting success stories such as Neotoma are at risk due to reliance on grant-based funding. Consequently, sustainable data infrastructure requires treating data contribution not as an obligation, but as a scholarly practice, and databases should be thought of not as products, but as commons

**Table 1 | The value of the samples and collections (sites) stored within three active palaeo databases in US dollars**

| DB | Samples (*n*) | Collections (*n*) | Sample value (US$) | Collections value (US$) | Total (US$) |
|---|---|---|---|---|---|
| PBDB | 1,653,699 | 240,405 | 248,054,850 | 721,215,000 | |
| GBDB | 580,049 | 217,969 | 87,007,350 | 653,907,000 | |
| Neotoma | 12,281,094 | 25,168 | 1,842,164,100 | 75,504,000 | |
| | | | 2,177,226,300 | 1,450,626,000 | 3,627,852,300 |

Conservative value estimates are taken from the valuation framework[69] and do not include collection and curation labour, storage, development, maintenance and institutional overhead, which are collectively more than double the presented estimates. The original valuation of Neotoma[69] has been expanded to include PBDB and GBDB. Samples refer to individual records, for example, species occurrence in PBDB. A collection refers to a grouping of samples—for example, a geographical site such as an outcrop in GBDB, or a field location in Neotoma.

sustained by collective stewardship. Volunteer labour in data contribution and backend development is often invisible and rarely credited, yet it is what has kept these long-lived databases active.

Databases achieving longevity exceeding 20 years tended to use one or more of three distinct strategies. First, some databases have relied on dedicated volunteer maintenance by one or two individuals with free or cheap institutional hosting support (for example, NSB). This solution can extend database longevity and sustainability through ties to the career of individuals but faces challenges when those individuals retire or shift positions. Second, some databases have enhanced their sustainability and achieved economies of scale by joining together and integrating into larger cooperative structures. For example, Neotoma was first formed as the union of FAUNMAP, the Latin American Pollen Database and other continental-scale databases, and new Constituent Databases continue to form and join Neotoma to leverage its data model and services[10,37]. Third, direct community-driven data contribution: PBDB has grown through primary data uploads from hundreds of volunteer contributors (see 'Curator review' in Supplementary Data). Within community initiatives, both the cooperative-database model and the direct-volunteer model leverage international research communities to build and grow their databases, while the first three strategies all rely on competitive grant funding to sustain and develop their data infrastructure. This community effort was also supported by the introduction of novel funding systems, such as the US National Science Foundation (NSF) Geoinformatics programme, which shifted its support for scientific databases from a traditional 3-year model to a development-dependent model. This new model includes a 3-year ramp-up stage for new resources, a primary database support stage lasting up to 10 years (divided into 3- to 4-year competitive awards) and a 3-year ramp-down stage. Database longevity is thus linked to both sustained community investment in volunteered time, experience and data contributions and to new funding models that support sustained, community-led database growth.

## Towards the third generation of palaeontological databases

We present here a series of actionable recommendations to address the existing structural and community challenges within the palaeontological and Earth science data landscape (Table 2). To address data fragmentation and structural redundancy in databasing effort, the immediate priority is to maximize the value of existing services while laying the groundwork for long-term solutions.

### Modular, interoperable and community-led data systems

The scientific community and governing bodies (for example, funders) must move away from the current trend of creating standalone databases that are not interoperable and either too small or too disconnected from their communities to achieve longer-term sustainability. Instead, they must design for integration and community engagement, to break the cycle of effort and loss. While broader challenges around data infrastructure are often shaped by political and institutional forces beyond the control of individual researchers, the scientific community can take meaningful action through improved data practices[64,80–82].

Examples such as Neotoma, NOW and PBDB, which have remained active for over 15 years and continue to serve global communities across disciplines, demonstrate the efficacy and resilience of collaborative stewardship[9,37,83]. Notable features of long-lived databases include international collaboration in data stewardship, critical community contributions by way of volunteered data, and efficient data ingestion (see 'Historical trends in Results' section; see also 'Curator review' in Supplementary Data). However, databases and related resources such as the Biodiversity Heritage Library (https://www.biodiversitylibrary.org/) remain vulnerable to 'extinction', such as through cyberattacks, but more commonly due to funding termination.

By prioritizing interoperability, modularity and close engagement between databases and their supporting communities, we can build a resilient and pluralistic community of data systems that safeguard multiple dimensions of scientific data and ensure its continued relevance to scientists, external stakeholders and the general public[66,84]. To this end, we recommend the transition to a decentralized modular data network (Fig. 3), where core components, such as those responsible for taxonomy, stratigraphy and specimen provenance, are built with a flexible scope and in a way that minimizes duplicative curatorial effort. In this vision, each part of the scientific community would be responsible for curating a specific area of data and knowledge (for example, age models and time inferences; stratigraphy and lithology; taxonomies and phylogenies; organismal abundance and occurrence; fossil morphologies; and ecological traits), and these modular, interconnected systems would integrate data and knowledge across these domains. This system would function by transitioning from the fragmented and uncoordinated data landscape (Fig. 3a) to pooled, pluralistic frameworks[66] (Fig. 3b). Pluralistic approaches to data pooling maintain domain independence and flexibility, permitting field-specific misalignment (for example, the unit differences in terminology and grouping seen between core-based micropalaeontology and global macrofossil biogeography in terms of spatial and temporal binning; Fig. 3b). Modules within this system serve as interlocking elements, offering researchers the foundation to develop extension structures necessary for addressing novel scientific questions within a broader, connected data landscape (Fig. 3b). For example, to answer questions about fossil biotic interactions (for example, BITE[85]), a new data structure is required, developed specifically to tie one or more biotic interactions and the organisms to a rock specimen. This novel database element would then be integrated with pre-existing core elements such as taxonomy, stratigraphy and geography (Fig. 3b), meaning the only new element to be constructed is the one that captures explicitly biotic interaction data. This approach saves time on database construction, reduces duplicative effort, ensures interoperability and safeguards against the loss of data from novel databases. In this way, the data from 'extinct' databases can be conserved and reintegrated, either by adding them to existing core modules or by creating new modules. The suggested solution mimics the general tendency of some corporations that move from large monolithic applications to interconnected microservices to meet the demands of scalability and a fast development cycle[84,86], and is particularly suited to scientific research that is globally distributed in nature[48–50,80].

## Table 2 | A roadmap to sustainable funding

| Action | Description |
|---|---|
| (1) Embed sustainability from inception | Design databases with modular architecture and interoperability in mind. Incorporate regional and linguistic equity in API development. This enables future integration into broader infrastructures and reduces redundancy, lowering long-term maintenance costs. |
| (2) Establish core infrastructure grants | Advocate for dedicated infrastructure funding schemes for domestic and international initiatives, distinct from research project grants, which support long-term maintenance, technical upgrades and data curation. Prioritize capacity building within the community in both database curation and database use. |
| (3) Develop cross-sector partnerships | Collaborate with museums, universities, government agencies and industry partners to co-invest in shared data resources. |
| (4) Quantify and communicate value | Systematically assess the scientific and economic value of databases to demonstrate return on investment and attract strategic funding. |
| (5) Adopt attribution standards | Promote data citation, DOI assignment and recognition mechanisms to incentivize community data contributions and support funding applications that highlight demonstrable use. |
| (6) Foster community governance | Create steering bodies or consortiums to coordinate long-term strategy, technical development and funding pipelines across institutions and borders. |

The community enthusiasm for developing shared resources and initiatives is evident in our data landscape. The proposed roadmap relies on structured communication of the value and importance of community-developed databased, while developing cross-sector relationships and expanding community buy-in.

To realize their full potential, third-generation databases must, whenever possible, maintain direct links to physical specimens and samples (for example, through the International Generic Sample Numbers; https://ev.igsn.org/), originators and users (for example, persistent identifier through ORCID; https://orcid.org/), and usage (for example, DATACITE for DOI mining; https://datacite.org/), while also improving linkages to other databases (see 'The near future—from databases to third-generation data systems' section). Museums, research institutes and public collections are a foundation of this system, providing crucial metadata that tie scientific conclusions based on digital data to the primary physical evidence[68,87,88]. Strengthening links between physical specimens and their digital representations will ensure long-term data accessibility, foster interdisciplinary research and empower the next generation of large-scale palaeontological analyses by upholding scientific transparency and rigour[48,65,89].

Developing application programming interfaces (APIs)—which enable one software program to request services or data from another without needing to know the other system's internal workings and that adhere to open science standards[19,90–92]—is crucial for ensuring seamless exchange of information between data systems, regardless of their underlying technologies. In addition, data harmonization tools[93] can streamline data integration and scientific workflows by automatically reconciling differences in data formats, units of measurement and terminologies. For example, the fossilpoll workflow[94] (hope-uib-bio.github.io/FOSSILPOL-website/en/index.html) pulls data from Neotoma, harmonizes the age-depth models and builds harmonized taxonomic names lists. These workflows create opportunities to distribute effort, allowing scientists outside the database leaders/curators to add value while establishing strong provenance between these downstream research analyses and the databases. Furthermore, such tools can leverage artificial-intelligence and machine-learning algorithms to help data stewards identify and merge duplicate records, standardize taxonomic names and align stratigraphic information, reducing the manual effort required for data integration. Because of the complexity of fossil data and the implicit knowledge often embedded in palaeontological datasets, we recommend that analytical and curatorial workflows use human-in-the-loop approaches rather than fully automated systems, to avoid 'garbage in, garbage out' situations and ensure accuracy and reliability.[26]

### Financial support

Using palaeontological data as an example, we propose a path forward for sustainable development, funding and stewardship to safeguard community-built scientific data systems for future generations. While we focus here on open digital resources for the democratization of science, investment in such resources must be accompanied by stronger linkages to, and explicit support for, museums and physical repositories[64,65,95–97].

Long-lived databases have been developed and maintained through a combination of sporadic funding, international support and unfunded volunteer/service work[61,69,98]. The persistence of these databases through all this financial precarity is a testament to their importance and the work of many scientists to keep them going. Investing in sustainable, modular data infrastructure not only enhances the longevity, accessibility and utility of scientific data, but also protects the immense financial and intellectual investment already made[61,98]. Funding is essential for ensuring that community-curated data continue to inform cutting-edge science well into the future.

Besides optimizing the use of already acquired funding, long-term sustainability hinges on moving beyond short-term, project-driven funding models (Tables 2 and 3) such as those offered by the NSF Geoinformatic programme model. Advocating for policy support at institutional, national and international levels is required to create an environment for these systems to thrive[97]. Network-level integration provides a means to ensure continued relevance, usability and return on investment beyond the end of a research project's funding cycle[69,98].

Engaging policymakers and funding agencies in discussions about the importance of Earth science and palaeontological community data networks can help to secure the necessary support and resources (for example, the USA's Geoscience Congressional Visit days). Core infrastructure funding, akin to utilities for the scientific community, should be secured through national and international bodies, ensuring that databases are treated as essential research infrastructure (for example, the Australia Data Research Commons (https://ardc.edu.au/) and the Chinese National Natural Science Fund Key Basic Research Infrastructure programme (nsfc.gov.cn/english/site_1/funding/E1/2024/06-12/364.html), which supports geo-data infrastructure). Within our proposed funding roadmap (Table 2 and Fig. 3), we recommend demonstrating the economic, societal and scientific value of open data through public–private partnerships and cost–benefit analyses, approaches already proven effective in initiatives such as Ozboneviz[97]. Ultimately, we recommend the establishment of a dedicated international non-profit organization, akin to the European Organization for Nuclear Research (CERN) or GBIF, which would advance the financial sustainability of the life and Earth science data landscape.

These two organizations, among others, provide a useful template for palaeontology and Earth systems science more broadly. The success of the GBIF, for example, lies in its clear governance, strategic coordination and stable funding model—elements that palaeontological and Earth science databases are yet to fully achieve (Fig. 3). GBIF operates as a community-governed, multinational consortium supported by member countries, each contributing financially and through data provision, underpinned by a robust strategic framework that ensures long-term stability, interoperability and open access[55,99,100]. Its structure—from local and national nodes to global coordination—promotes accountability and sustained collaboration, while its standards (for example, Darwin Core[56]) have become a foundation for data integration across the life sciences. The proposed framework for palaeontology (Table 2 and Fig. 3) builds upon this foundation while simultaneously

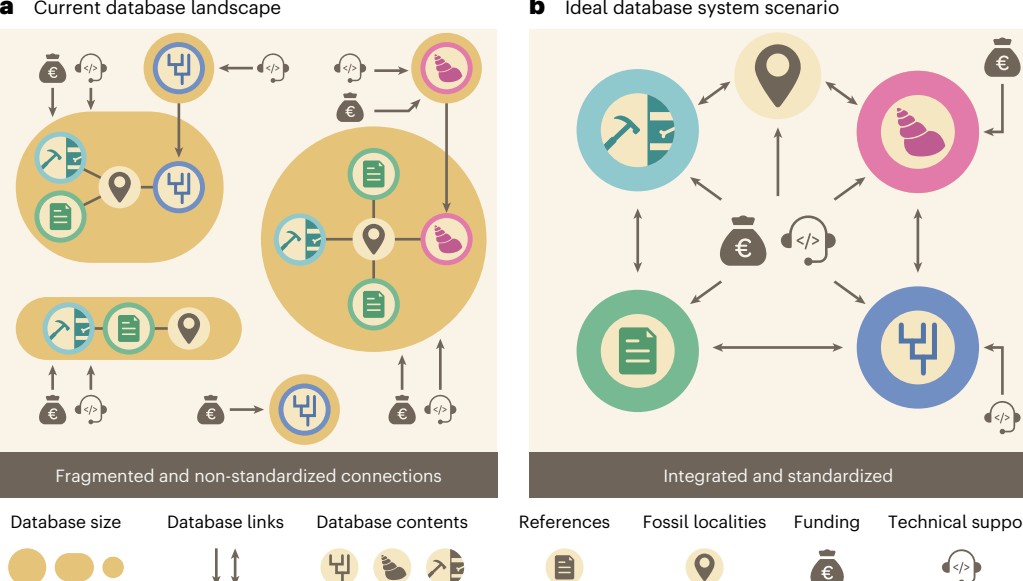

**a** Current database landscape

**b** Ideal database system scenario

Fragmented and non-standardized connections

Integrated and standardized

Database size Database links Database contents References Fossil localities Funding Technical support

**Fig. 3 | Graphical representation of the current database landscape and a possible idealized scenario for the structure of the palaeobiological database landscape. a**, The current data landscape consists of disparate databases of varying size, scope and resolutions that are connected through limited links. This means that core elements (such as fossil localities) each have independent and therefore repeated solutions for each database. **b**, An idealized database system consists of a decentralized network of interconnected independent subsystems (nodes) that benefit from collective standards and potential sharing of financial and technical support. Core data elements, such as fossil localities, are accessible as a standardized module that each database uses when developing domain-specific data structures (for example, for phylogenetic matrices or stratigraphy). A central support system, without compromising data sovereignty, can be put in place to decrease funding volatility and ensure network-level standards and integrity. The maintainers of individual databases can then apply for specific external grants and technical support to develop tailored data solutions to address novel scientific questions. Credit: Science Graphic Design.

acknowledging that pluralist approaches to establishing and formalizing collaboration are essential. Stable financial support is required to bridge the domain-specific knowledge and data structures required to hold information from the multiple subdisciplines of palaeontology (for example, palaeobiology, biostratigraphy, ichnology and palaeobotany) and related Earth sciences (for example, sedimentology, geodynamics, geochemistry and climatology). This inclusivity ensures that the framework not only supports technical interoperability but also fosters equitable participation and long-term sustainability across the full spectrum of palaeontological and Earth science research (Fig. 3). For example, a recent review of geochemical databases highlights similar trends in data lifecycles, while highlighting that geochemical data require tailormade data structures to host and develop them[61]. In understanding the data requirement, and if led by those creating and using the data, a GBIF-like framework (that is, formalized international partnerships, strategic cooperative leadership, modular infrastructure and clear attribution systems) can secure sustainable data management, enhance interoperability and ensure the long-term preservation and growth of palaeontology and Earth science's collective digital resources.

## Community governance and goals

We propose a phased, community-guided transition towards a sustainable, interconnected, and explicitly modular data infrastructure—one that is grounded in equitable practice and ensures proper attribution[40,64,91,101–103]. As artificial intelligence, large-scale web scraping and automated data aggregation become increasingly common tools, the palaeontological community must actively shape how its openly accessible data are structured, cited and reused[25,40,98,102]. A modular and well-governed framework will allow us to respond nimbly to these technological developments while preserving the integrity and provenance of our data. Central to this vision is strong, inclusive community governance—led by the researchers, data stewards and institutions who

know the data and needs of the researchers best[66]. By harmonizing efforts and redistributing responsibilities through open consultation, we can build an equitable and future-ready infrastructure that supports both innovation and accountability in palaeoscience.

Promising steps are already underway. Initiatives such as the ARC Centre of Excellence for Australian Biodiversity and Heritage (CABAH; epicaustralia.org.au) exemplify how community-led, transdisciplinary frameworks can successfully balance Indigenous knowledge systems, biodiversity and palaeodiversity data, and open infrastructure. In 2023, CABAH produced 127 journal articles and welcomed over 60,000 attendees to its public programmes and events. CABAH's approach is collaborative, bringing together researchers, Indigenous communities, industry and policy partners. This momentum is furthered by ensuring that decisions around standards, attribution and data validation are made through inclusive consultation with a broad cross-section of the community, including historically underrepresented groups and the global majority.

Community buy-in for data attribution and validation will facilitate community trust in open data resources[90–92]. True integration goes beyond technical aspects and requires active collaboration between scientists and technical experts from varied disciplines (Table 3 and Fig. 3). Establishing interdisciplinary data standards, training programmes, research teams and projects can facilitate this collaboration (Table 2). Through this effort, we can develop common research frameworks and questions to guide data integration efforts, aligning the objectives of different disciplines[63].

## Conclusions

Palaeontological data systems are critical resources for the advancement of Earth system research and the training and development of Earth scientists. By committing to the development and maintenance of decentralized, interconnected, modular data systems, we can address pressing questions about the history of life on Earth, ensure the

**Table 3 | Recommendations for the sustainable development of community-developed data resources and the related benefits derived from their implementation**

| Recommendation | Stakeholders | Details | Benefits |
|---|---|---|---|
| (A) Incentivize data contributions | Researchers, data curators, database developers, policymakers | Create systems (and a scientific culture) for increased acknowledgement, attribution and citation for data contributions. | 4, 5, 6 |
| (B) Establish a framework for data integration | Researchers, data curators, database developers, funders | Develop a standardized framework for integrating diverse Earth system databases, ensuring interoperability and data quality transparency. | 1, 2, 5, 6 |
| (C) Secure sustainable funding | Researchers, database developers, funders, policymakers, institutions | Advocate for dedicated funding streams to support the development, maintenance and enhancement of modular data systems. | All |
| (D) Promote open science practices | All | Encourage the adoption of open science practices, including open data, open-access publications and collaborative research initiatives. | All |
| (E) Invest in technology and innovation | Funders, policymakers, institutions | Leverage technological advancements to enhance data integration, analysis and visualization capabilities. | 1, 2, 6 |
| (F) Build and foster global collaborations | Researchers, funders, policymakers, institutions | International collaborations and partnerships create a comprehensive and diverse global network of palaeontological data. | 2, 4, 6 |
| (G) Ensure ethical and legal compliance | All | Addressing ethical and legal considerations, including data privacy, security and intellectual property rights, ensures responsible data management and sharing. | 1, 4, 6, 7 |
| (H) Advocate for policy support | All | Advocating for policy support at institutional, national and international levels is required to create an environment for these systems to thrive. | All |

The benefits are rigour and reliability (1), ability to address new questions (2), faster and more inclusive dissemination of knowledge (3), broader participation in research (4), effective use of resources (5), improved performance research tasks (6) and open publication for public benefit (7; see Supplementary Table 3 for expansion and descriptions).

longevity of our shared resources and create a more interconnected scientific community. This effort is already underway, building on the success of first- and second-generation data systems that have advanced our understanding and technical capacity. Developing integrated support systems will protect, sustain and enhance these valuable community-driven data resources. Together, these recommendations align structural reforms with scientific needs and community values. The path forward requires a collective effort, sustained funding and a commitment to collaboration, ensuring that palaeontological data remain valuable resources for future generations.

# Methods
## Database survey
To assess the temporal dynamics and sustainability of palaeontological databases, we systematically searched Web of Science and Google Scholar between November 2024 and March 2025. Search terms combined multilingual instances of 'database' (Supplementary Table 4) with 'palaeontology', 'geology', 'fossil' and 'Earth science'. Web of Science searches were restricted to 'Physical', 'Chemical & Earth Sciences' and 'Life Sciences' categories. Languages were selected on the basis of distribution by official or co-official status: English, Spanish, Arabic and French, following country counts from the South Australian Government[104]. Additional major languages (for example, Mandarin) were searched for up to five pages, with searches terminated when no new non-governmental databases were identified.

We inspected the first 10 result pages per aggregator (100 results for Google Scholar; 250 for Web of Science). Each result was examined to distinguish presentations of new databases from studies citing existing databases. Results were recorded following standardized definitions (Supplementary Table 5).

## Temporal and funding data
Inception or start date (Supplementary Table 5) was defined as the year a database became publicly available, determined by the earliest of associated publication date or website launch. End date (last reference/update; Supplementary Table 5) was the most recent documented update, identified hierarchically from: (1) database website update information, (2) publications documenting database state, (3) most recent citation in scientific literature or (4) last confirmed year of public accessibility. Databases with identical start and end dates were included in diversity metrics but excluded from longevity and

extinction analyses as they represent point occurrences rather than temporal spans. Funding information was recorded from database websites or associated publications when available. Records lacking start or end dates were omitted from analyses.

## Database analysis
We identified 171 palaeontological and Earth science databases. After removing governmental databases (see rationale in Supplementary Table 5), 125 remained, of which 118 met the inclusion criteria (see 'Richness' in Supplementary Data). Summary statistics on database duration excluded same-year databases ($n$ = 30), yielding 88 databases with temporal ranges (Supplementary Tables 6 and 7). An additional analysis excluded the top 15% longest-lived databases (approximately 25 years; Supplementary Table 6) to examine diversity dynamics, because the diversity and number of databases through time and associated changes affect the data landscape and its stability (Fig. 2), representative of typical community-maintained databases.

The per-capita extinction and origination rates were analysed using a rolling mean of year-to-year database activity, while the sampled-in-bin diversity used an extended decadal time series to account for boundary conditions.

To mitigate edge effects, we extended end dates of active databases to 2027 and truncated analyses at 2024. This approach addresses the pull-of-the-recent bias affecting terminal rates. The time series start was extended to the 1970s to include early static datasets that predate digital database proliferation.

Analyses were conducted in RStudio (4.5.0)[105,106] using DivDyn[107]. To address the question of database diversity dynamics, we calculated:
- Richness: total number of databases active within a time (divSIB);
- Diversity by duration: distribution of databases by years active (divRT);
- Origination rate: the rate at which new databases are established per year (2-year rolling mean; PC:oriPC);
- Extinction rate: the rate at which databases cease activity per year (2-year rolling mean; PC: extPC).

Rolling means used a 2-year window to smooth interannual variation. All raw data, including point occurrences (same-year databases), were included in rolling mean calculations to capture complete database origination dynamics. The following metrics

were considered in both raw and rolling mean treatments for origination, extinction and diversity used in Fig. 2: sampled-in-bin diversity, range-through diversity, per-capita extinction and per-capita origination. A full list of 12 metrics (Supplementary Table 8) was calculated.

### Author survey on database maintainers, curators and data contributors

The authors of this paper, who were also database maintainers and/or developers, volunteered information about the backend, data volume and support structures (Supplementary Table 2). The authors presented data across 68 categories, including 'History and funding management' (7 categories), 'Scope' (3 categories), 'Software and maintenance' (16 categories), 'Data contained' (5 categories), and 'Entity feature' coverage (37 categories; see 'Curator review' in Supplementary Data). These descriptions informed benefits and recommendations (Tables 2 and 3) and present a clear synthesis of the variability in database structure and maintenance. The provided database ages were incorporated into the database survey (Fig. 2), in addition to funding and technical support summaries.

The citation count for each database was also requested from the database maintainers. PBDB was selected for full consideration, while Neotoma, the Geobiology Database, Triton, Neptune and NOW are present for completeness (see 'Palaeodatabase publication products' in Supplementary Data). The published literature (>1,800 papers) that cited PBDB as a data contributor were each tagged using 15 categories (palaeobiogeography, diversity, taxonomy, morphology, phylogeny, palaeoecology, environment, taphonomy, palaeoclimate, conservation, geochemistry, sedimentology, sedimentology, stratigraphy, evolution and other) to capture the diversity of topics PBDB data are used for. Owing to citation practices, the number of formal citations gained by PBDB is notably lower than the citing literature or 'mentions' the database when querying aggregators (>34,000 from Google Scholar, February 2025).

### Methods for financial valuation

We used a financial valuation framework[69] on the data volume that was provided either by the database maintainers (see 'Curator review' in Supplementary Data) or the most recent version of the database as of June 2025. The rationale valuation centres on the cost of replacing the data assuming only labour, expertise and institutional overhead are required[69]. The rationale also assumes that the data can be collected again, that the sites are still accessible, and that equal quality specimens can be obtained. Within palaeontology and Earth sciences, this is often not the case. We elected to focus on only two of the options: sample value (US$150) and site value (US$3,000).

Additional costs for publication, data hosting, hiring database maintainers and developers, and curatorial labour were not included in the valuation (and also were not listed in ref. 69 the valuation formula).

### Reporting summary

Further information on research design is available in the Nature Portfolio Reporting Summary linked to this article.

## Data availability

All data generated for the author survey, publication, products and analysis are available within the Supplementary Information and via Zenodo at https://doi.org/10.5281/zenodo.17828000 (ref. 108). Source data are provided with this paper.

## Code availability

All code and required data are available via CodeOcean at https://doi.org/10.24433/CO.1586965.v1 (ref. 109). All analyses were conducted using publicly available R packages, and the links have been provided in the Methods.

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

## Acknowledgements

This Analysis was written as part of the 'Integrated Record of Ancient Life' (IRAL) working group with support from the Paleosynthesis Project. E.M.Do. thanks C. Piper, Western Australia Biodiversity Information Office and Dandjoo Biodiversity Data Repository, for her insight into public policy, curation and physical collections. Funding sources for individual investigators include the following: E. M. Dowding, D.D., W.K., Á.T.K. and B.S.—Volkswagen Stiftung, Az 96 796; E. M. Dunne—FAU Emerging Talents Initiative; E.E.S.—Leverhulme Prize, NERC grant NE/V011405/1, NERC grant NE/C001340/1; K.D.B—I.3.4 Action of the Excellence Initiative – Research University Programme at the University of Warsaw (project: PARADIVE); L.N.— National Natural Science Foundation of China (42372039); J. A.

Sessa—NSF 1928362; M.D.U.—NSF EAR 1948831; J.W.W.—NSF 2410961. Figures 1 and 3 were designed with M. Kouvari and N. M. Morales Garcia from Science Graphic Design (sciencegraphicdesign.com).

## Author contributions

E. M. Dowding conceived and designed the study. E. M. Dunne and Á.T.K. acquired funding and framed the workshop. E. M. Dowding and K.C. collected the data for database diversity survey analyses. K.S.C., K.D.B., D.D., S.M.E., S.F., W.K., K.L., L.H.L., H.L., L.N., S.E.P., J.R., E.E.S., J. A. Sessa, J. A. Smith, M.D.U., J.W.W. and Á.T.K. provided technical information about database curation and software. E. M. Dowding led the writing with E. M. Dunne, J. A. Smith and J.W.W. All authors contributed reviewed, edited and approved the final version.

## Funding

## Competing interests

The authors declare no competing interests.

## Additional information

**Correspondence and requests for materials** should be addressed to Elizabeth M. Dowding, Emma M. Dunne or Ádám T. Kocsis.

[1]GeoZentrum Nordbayern, Friedrich-Alexander-Universität Erlangen-Nürnberg, Erlangen, Germany. [2]School of Natural Sciences, Geology, Trinity College Dublin, Dublin, Ireland. [3]Natural History Museum, London, UK. [4]Institute of Evolutionary Biology, Faculty of Biology, University of Warsaw, Warsaw, Poland. [5]Department of Paleobiology, National Museum of Natural History, Smithsonian Institution, Washington, DC, USA. [6]Department of Integrative Biology, University of California, Berkeley, Berkeley, CA, USA. [7]Smithsonian Tropical Research Institute, Panama City, Panama. [8]Natural Sciences Unit, Finnish Museum of Natural History, Helsinki, Finland. [9]Natural History Museum, University of Oslo, Oslo, Norway. [10]Centre for Planetary Habitability, Department of Geosciences, University of Oslo, Oslo, Norway. [11]State Key Laboratory of Paleobiology and Stratigraphy, Nanjing Institute of Geology and Palaeontology, Chinese Academy of Sciences, Nanjing, China. [12]Department of Geoscience, University of Wisconsin-Madison, Madison, WI, USA. [13]Museum für Naturkunde, Leibniz-Institut für Evolutions- und Biodiversitätsforschung, Berlin, Germany. [14]Department of Earth Sciences, University of Oxford, Oxford, UK. [15]Academy of Natural Sciences of Drexel University, Philadelphia, PA, USA. [16]Department of Earth and Environmental Sciences, University of Minnesota Duluth, Duluth, MN, USA. [17]Department of Atmospheric, Oceanic, and Earth Sciences, George Mason University, Fairfax, VA, USA. [18]Department of Geography, University of Wisconsin-Madison, Madison, WI, USA. ✉e-mail: dowding.e.m@gmail.com; dunne.emma.m@gmail.com; adam.kocsis@fau.de

# Reporting Summary

## Statistics

For all statistical analyses, confirm that the following items are present in the figure legend, table legend, main text, or Methods section.

| n/a | Confirmed | |
|---|---|---|
| ☐ | ☒ | The exact sample size (*n*) for each experimental group/condition, given as a discrete number and unit of measurement |
| ☐ | ☒ | A statement on whether measurements were taken from distinct samples or whether the same sample was measured repeatedly |
| ☐ | ☒ | The statistical test(s) used AND whether they are one- or two-sided<br>*Only common tests should be described solely by name; describe more complex techniques in the Methods section.* |
| ☒ | ☐ | A description of all covariates tested |
| ☐ | ☒ | A description of any assumptions or corrections, such as tests of normality and adjustment for multiple comparisons |
| ☐ | ☒ | A full description of the statistical parameters including central tendency (e.g. means) or other basic estimates (e.g. regression coefficient) AND variation (e.g. standard deviation) or associated estimates of uncertainty (e.g. confidence intervals) |
| ☒ | ☐ | For null hypothesis testing, the test statistic (e.g. *F*, *t*, *r*) with confidence intervals, effect sizes, degrees of freedom and *P* value noted<br>*Give P values as exact values whenever suitable.* |
| ☒ | ☐ | For Bayesian analysis, information on the choice of priors and Markov chain Monte Carlo settings |
| ☒ | ☐ | For hierarchical and complex designs, identification of the appropriate level for tests and full reporting of outcomes |
| ☒ | ☐ | Estimates of effect sizes (e.g. Cohen's *d*, Pearson's *r*), indicating how they were calculated |

*Our web collection on statistics for biologists contains articles on many of the points above.*

## Software and code

Policy information about availability of computer code

| Data collection | Curator review data was collected from a survey of the author team and involved reporting information as of Nov 2024 about the back-end development, data volume, and funding of the databases the author represented. Publication products was supplied by the curators who collated formal citations of their work (the database publication) in other published literature; these were then assigned keywords on broad topics within palaeontology. Database diversity dynamics data was collected from a web survey of databases first publication, most recent publication, and whether the database was still actively available and maintained. These were conducted in the last quarter of 2024 and the first of 2025.<br>Data and code are available at CodeOcean, Zenodo and GIThub [https://github.com/dowdingem/IRAL]. The study has no restriction on data availability.<br><br>All data, code and supplementary material:<br>Dowding, E. M., et al. Fossils for Future and the billion dollar case for palaeontology's digital infrastructure. [Dataset] Zenodo. (2025). https://doi.org/10.5281/zenodo.17828000<br><br>Code and relevant data:<br>Dowding, E. M., et al. Fossils for Future: the billion dollar case for palaeontology's digital infrastructure. [Codebase] CodeOcean. (2025) https://doi.org/10.24433/CO.1586965.v1 |
|---|---|
| Data analysis | Analysis was run in R v4.5.0 using the DivDyn Rpackage (Kocsis et al 2019) which contains metrics for richness, origination, extinction, and range through diversity amongst others. The information is available on CodeOcean and the GIThub links provided. In the time series analyses, |

the durations of active and recent <5 years old) databases were extended into the future and the analytical frame was truncated at 2024 to address edge effects.

For manuscripts utilizing custom algorithms or software that are central to the research but not yet described in published literature, software must be made available to editors and reviewers. We strongly encourage code deposition in a community repository (e.g. GitHub). See the Nature Portfolio guidelines for submitting code & software for further information.

## Data

Policy information about availability of data

All manuscripts must include a data availability statement. This statement should provide the following information, where applicable:

- Accession codes, unique identifiers, or web links for publicly available datasets
- A description of any restrictions on data availability
- For clinical datasets or third party data, please ensure that the statement adheres to our policy

> All data generated for the author survey, publication, products, and analysis are available within the Supplementary Material and the stable Zenodo repository Dowding, E. M., et al. Fossils for Future and the billion dollar case for palaeontology's digital infrastructure. [Dataset] Zenodo. (2025). https://doi.org/10.5281/zenodo.17828000
> All code and required data are available through CodeOcean. All analyses were conducted using publicly available R packages, and the links have been provided in Methods.

## Research involving human participants, their data, or biological material

Policy information about studies with human participants or human data. See also policy information about sex, gender (identity/presentation), and sexual orientation and race, ethnicity and racism.

| | |
|---|---|
| Reporting on sex and gender | *Use the terms sex (biological attribute) and gender (shaped by social and cultural circumstances) carefully in order to avoid confusing both terms. Indicate if findings apply to only one sex or gender; describe whether sex and gender were considered in study design; whether sex and/or gender was determined based on self-reporting or assigned and methods used.*<br>*Provide in the source data disaggregated sex and gender data, where this information has been collected, and if consent has been obtained for sharing of individual-level data; provide overall numbers in this Reporting Summary. Please state if this information has not been collected.*<br>*Report sex- and gender-based analyses where performed, justify reasons for lack of sex- and gender-based analysis.* |
| Reporting on race, ethnicity, or other socially relevant groupings | *Please specify the socially constructed or socially relevant categorization variable(s) used in your manuscript and explain why they were used. Please note that such variables should not be used as proxies for other socially constructed/relevant variables (for example, race or ethnicity should not be used as a proxy for socioeconomic status).*<br>*Provide clear definitions of the relevant terms used, how they were provided (by the participants/respondents, the researchers, or third parties), and the method(s) used to classify people into the different categories (e.g. self-report, census or administrative data, social media data, etc.)*<br>*Please provide details about how you controlled for confounding variables in your analyses.* |
| Population characteristics | *Describe the covariate-relevant population characteristics of the human research participants (e.g. age, genotypic information, past and current diagnosis and treatment categories). If you filled out the behavioural & social sciences study design questions and have nothing to add here, write "See above."* |
| Recruitment | *Describe how participants were recruited. Outline any potential self-selection bias or other biases that may be present and how these are likely to impact results.* |
| Ethics oversight | *Identify the organization(s) that approved the study protocol.* |

Note that full information on the approval of the study protocol must also be provided in the manuscript.

# Field-specific reporting

Please select the one below that is the best fit for your research. If you are not sure, read the appropriate sections before making your selection.

☐ Life sciences  ☐ Behavioural & social sciences  ☒ Ecological, evolutionary & environmental sciences

For a reference copy of the document with all sections, see nature.com/documents/nr-reporting-summary-flat.pdf

# Ecological, evolutionary & environmental sciences study design

All studies must disclose on these points even when the disclosure is negative.

| | |
|---|---|
| Study description | The study focuses on the diversity trends of community run Earth science databases, centred around Palaeontology. The study is both qualitative and quantitative, descriptions of the databases both in terms of history, publication product, and technical development were provided by the curators through survey of the author team. Quantitative analysis of the diversity dynamics (richness, origination, cumulative sum, and extinction) were run using first and last appearance data of databases that were open access and community run. Both were completed to show the diversity of databases, both in terms of their proliferation, but also |

into terms of the variation in their volume, and technical development (back end construction). These together for the basis for conclusions and recommendations.

| | |
|---|---|
| Research sample | Open access data bases were determined by both self description and the ability of a general member of the public to gain access to the data on their own or by request to the data administrator. Community run databases were identified by negative conditions, i.e. by not being managed by industry or government. |
| Sampling strategy | No sample size estimates were required as the analysis focuses on trends in time series and does no significance testing. Sampling strategy was by survey of the authorship team who represent the community run databases, and the diversity dynamics was by consistent search of free online aggregators using search terms in languages with the broadest geographic spread. |
| Data collection | Data was collected by Dowding using surveys which consisted both of a questionnaire and a spreadsheet for database curators to enter information about the database they maintain. |
| Timing and spatial scale | The information on databases first appearance (publication) and last point of activity (update publication or statement) was from the first appearance of large scale compilations that were published to 2024. Databases that fit the criteria of being open access and community run globally were included. |
| Data exclusions | Databases that were not open access or were governmental/industry maintained were excluded. Failing either or both of these categories resulted in exclusion from analysis. |
| Reproducibility | The data survey techniques are available in the supplementary information in addition to the raw data, clean data, and code. R ver. 4.5.0 |
| Randomization | N/A |
| Blinding | Blinding was not required as there are no experimental constraints and information from the curators were self reported and they are named authors. |

Did the study involve field work?  ☐ Yes  ☒ No

# Reporting for specific materials, systems and methods

We require information from authors about some types of materials, experimental systems and methods used in many studies. Here, indicate whether each material, system or method listed is relevant to your study. If you are not sure if a list item applies to your research, read the appropriate section before selecting a response.

## Materials & experimental systems

| n/a | Involved in the study |
|---|---|
| ☒ ☐ | Antibodies |
| ☒ ☐ | Eukaryotic cell lines |
| ☒ ☐ | Palaeontology and archaeology |
| ☒ ☐ | Animals and other organisms |
| ☒ ☐ | Clinical data |
| ☒ ☐ | Dual use research of concern |
| ☒ ☐ | Plants |

## Methods

| n/a | Involved in the study |
|---|---|
| ☒ ☐ | ChIP-seq |
| ☒ ☐ | Flow cytometry |
| ☒ ☐ | MRI-based neuroimaging |

## Plants

| | |
|---|---|
| Seed stocks | *Report on the source of all seed stocks or other plant material used. If applicable, state the seed stock centre and catalogue number. If plant specimens were collected from the field, describe the collection location, date and sampling procedures.* |
| Novel plant genotypes | *Describe the methods by which all novel plant genotypes were produced. This includes those generated by transgenic approaches, gene editing, chemical/radiation-based mutagenesis and hybridization. For transgenic lines, describe the transformation method, the number of independent lines analyzed and the generation upon which experiments were performed. For gene-edited lines, describe the editor used, the endogenous sequence targeted for editing, the targeting guide RNA sequence (if applicable) and how the editor was applied.* |
| Authentication | *Describe any authentication procedures for each seed stock used or novel genotype generated. Describe any experiments used to assess the effect of a mutation and, where applicable, how potential secondary effects (e.g. second site T-DNA insertions, mosiacism, off-target gene editing) were examined.* |

