## [Peer Review File · Nature Ecology & Evolution]

The billion dollar case for sustaining palaeontology's digital databases

Corresponding Author: Dr Elizabeth Dowding

Version 0:

Decision Letter:

14th October 2025

Dear Dr Dowding,

Your Review entitled "Fossils for Future: the billion-dollar case for palaeontology's digital infrastructure" has now been seen by three reviewers. You will see from their attached comments that there is lots of enthusiasm for the piece, but they have raised points that need to be addressed before we can make a decision on publication. We will need to consider your response to these concerns in the form of a revised manuscript, accompanied by a separate list to explain your revisions. Additionally, given the emphasis (and praise) from the reviewers on the data analysis and methods, we feel that the paper does not conform to our Review/Perspective format, and would better suit our 'Analysis' format, which combines review content with substantive data analysis and includes a separate methods section (as requested by the reviewers here). It's also fine to integrate all the references into the main reference list rather than partitioning some of them into the SI. When revising as an Analysis, please could you be more explicit about the data analysis and results in the abstract. And while the synthesis emerges as a valuable part of the paper, we're not sure that this really qualifies as a formal meta-analysis (which would require a PRISMA chart and other formal reporting), so I would suggest retitling those aspects. If you have any questions about the Analysis format as you revise, please don't hesitate to contact me by email, but you could also take a look at papers that we have previously published as Analyses to get an idea of the style. <https://www.nature.com/natecolevol/articles?type=analysis>

We very much hope that you will be able to address these comments of our reviewers and would like to invite you to revise your manuscript accordingly.

When you are ready, please use the link below to submit the revised version (with the changes clearly marked).

Link Redacted

- include a point-by-point response to any editorial suggestions and to our reviewers. Please include your response to the editorial suggestions in your cover letter, and please upload your response to the reviewers as a separate document.

- ensure it complies with our format requirements for Review articles as set out in our guide to authors at <https://www.nature.com/natecolevol/content>

- state in a cover note the length of the text, methods and legends; the number of references and the number of display items (figures, tables and uncaptioned molecular structures).

Please ensure that all correspondence is marked with your Nature Ecology & Evolution reference number in the subject line.

Nature Ecology & Evolution is committed to improving transparency in authorship. As part of our efforts in this direction, we are now requesting that all authors identified as 'corresponding author' on published papers create and link their Open Researcher and Contributor Identifier (ORCID) with their account on the Manuscript Tracking System (MTS), prior to acceptance. ORCID helps the scientific community achieve unambiguous attribution of all scholarly contributions. You can create and link your ORCID from the home page of the MTS by clicking on 'Modify my Springer Nature account'. For more information please visit please visit <http://www.springernature.com/orcid>.

I would appreciate it if you could tell me if you think you will be able to submit a revised manuscript, and also the likely timescale.

I look forward to hearing from you soon.

[redacted]

Reviewer Comments:

Reviewer #1 (Remarks to the Author):

Although I don't work in paleontology, I have dealt with building and integrating large databases in other contexts, and I found this work to be an interesting pitch for advancing database infrastructure. In the sub disciplines I am familiar with, probably the exact same narrative could be made about generations 1-3 identified by the authors. The manuscript has a clear and concise narrative and seems quite polished. Just a few random thoughts came to mind:

Is it worth mentioning the need for redundancy in skills across the community? Modularity partly addressing this by reducing the diversity of infrastructures that database developers need to understand, and community organized databases make this more feasible by distributing the effort across multiple research groups, but is it worth addressing the need for redundancy explicitly?

I like the back of the envelope calculation for value; provides a nice teaser for the title and is easy to understand in the text.

I like the specific mention of the cyberattack, as it really isn't something we tend to account for in database development and could be increasingly important, particularly as the value of the database grows.

Are there any more attributes of databases that have gone extinct besides 5 year funding cycle lifespans? Such attributes might help identify other strategies to enhance 3rd generation efforts

I wouldn't mind seeing a few more explicit examples of insights gained as a result of aggregating paleo data to make the value more tangible and address more specifically what we have gotten for \$3 billion. E.g., could you make the case that entire sub disciplines of paleontology have emerged, or at least been enabled, from connecting databases?

It seems like there's some successes and guidance to be borrowed from genbank and GBIF - what might those be for paleo data?

Reviewer #2 (Remarks to the Author):

Dowding et al. present a review of the data landscape for palaeontology, and derive recommendations to ensure continued relevance and future sustainability of these data systems. Given the rise in 'big data' and machine learning approaches across all sciences, data literacy and awareness of our global data infrastructure are becoming essential to all researchers. This review, therefore, is highly topical and very relevant to the audience of Nature Ecology & Evolution. However, I have a few concerns/suggestions, especially regarding the first half of the manuscript. I recommend publication of this article in Nature Ecology & Evolution following minor revisions.

General comments:

1. The introduction and first half of Section 2 seem overly simplified and general to me. Whilst the second half of the manuscript is very well written and has a clear story line, these first few pages lack that same focus: key topics in palaeontology and data science are introduced, but the connection is not always clear and I often found myself wanting a slightly more detailed explanation of specific concepts. I would recommend shortening the general part of the introduction to only the most relevant topics, and instead giving a few more details and examples on the concepts that are introduced and how they connect.
2. In contrast, Section 3, i.e. the new results presented in this article, seems a little too well summarised and a lot of the detail of the database meta-analysis is 'hidden' in the supplementary materials. I understand that the journal format does not allow for all methodological detail to be included in the main text. However, at present, this section came across as fairly cursory, back-of-the-envelope type calculations; and only after looking through all supplementary tables did the impressive scale of the work become apparent to me. If possible, I would recommend moving a few more details from the supplement into the main text here, perhaps at the expense of Sections 1 and 2.
3. Your results on database extinction rates are very sobering. You already give multiple reasons of why most data systems are set up to fail in the short- to mid-term. Can you say more about the success strategies of what enabled the few long-lived data systems to persist? You hint at possible strategies in Section 2 and in your recommendations, but I was also hoping to see an analysis of what set the long-lived systems apart from the others (and what they have in common). This information is already contained within the supplementary 'CuratorReview.xlsx', I believe, but never explicitly stated. Another aspect you only hint at is international cooperation, governance and funding: all successful data systems in other disciplines have some level of internationality in common – I think this would also be useful to add to your discussion.

More detailed comments are appended below that I hope will help to address these general points.

Kind regards,
Marthe Klöcking

Detailed Comments

Section 1, first paragraph, line 10: missing opening parenthesis in reference to the Glossary. This sentence (lines 10-15) is very long and a bit hard to follow.

Second paragraph, line 1: I would suggest the word 'archives' rather than 'repositories' here, see definitions in the CODATA RDM Terminology: <https://vocabs.ardc.edu.au/viewByld/685>.

Third paragraph, lines 5-6: replace 'databases and systems' with 'data systems'? Please explain what is meant by 'diversity dynamics'

Section 2.1, last paragraph, last sentence: 'provide evidence OF provenance'? This sentence is a bit unclear and could benefit from more detail/nuance. I can see two separate issues here: 1) the risk of erroneous data records in secondary databases when only using processed data; 2) the question of attribution and importance of preserving a record of data sources and provenance.

Section 2.2.1, line 4: missing closing parenthesis after 'Sepkoski, 1982'

Section 2.2.2, third paragraph: the list of example databases is a bit hard to follow, I would recommend separating examples with a semi-colon or similar for clarity.

Section 2.3, first paragraph, lines 4-7: sentence a bit hard to follow, and very abstract. Can you give an example of where modular design has facilitated interoperability?

Second paragraph, lines 5-9: what exactly are you referring to with 'inequities'? Data access, data coverage and bias in distribution, etc ...? Can you give an example?

Section 3.2, second paragraph, lines 6-8: do these or other systems in your analysis give any insights on joint, international governance? In my opinion this is one of the major hurdles to database sustainability: our science is global, but the funding tends to be local or national (and competitive). I think you need to comment on whether the successful databases in palaeontology are big players supported by individual, currently well-off nations (that dominate the rest of the world) or if there are international cooperations. GBIF is a fantastic example, and there may be others?

I also wonder if your discussion (either here or later) might benefit from more of a comparison with other disciplines. These are big questions that are being discussed by almost all data-driven research fields; we can definitely learn from each other, and your conclusions and recommendations are echoed in other disciplines. I am biased, of course, but Klöcking et al. (2025) give a similar review of the geochemical data landscape, including an estimate of the value of data systems, recommendations for the future and a comparison with success in other disciplines such as seismology and crystallography, which might be helpful as a starting point.

Klöcking, M., Lehnert, K. A., & Wyborn, L. (2025). Geochemical databases. In *Treatise on Geochemistry* (pp. 97–135). Elsevier. <https://doi.org/10.1016/b978-0-323-99762-1.00123-6>

Figure 2: perhaps I missed it, but please explain (again) what you mean by 'database richness', especially in comparison to 'diversity' (i.e. number)

Section 4.2: I think somewhere in this section you need to comment on how the three databases that are older than 25 years have managed to survive so long. What are their business model(s) and why were they successful?

Third paragraph, line 6 onward: NFDI4Earth do not currently provide basic funding for data infrastructure (only for specific projects that advance interoperability), and its long-term existence is still uncertain. Perhaps better examples would be ARDC (<https://ardc.edu.au/>), AuScope (<https://www.auscope.org.au/>) and EarthScope (<https://www.earthscope.org/>); maybe also EPOS (<https://www.epos-eu.org/>)?

Line 9 onwards (and Table 2): You mostly discuss national funding here: how can we move to more international governance and sustainability?

Section 4.3, third paragraph: not Figure 3 was missing from the manuscript files I had access to.

Supplementary Materials:

- Table 1 (Glossary): are you aware of the CODATA RDM terminology (<https://codata.org/initiatives/data-science-and-stewardship/rdm-terminology-wg/>; <https://vocabs.ardc.edu.au/viewByld/685>)? This glossary contains some widely used terminology, I wonder if it is worthwhile adopting existing definitions.

- Supp-Table 3 (Publication products): formatting seems inconsistent between the different table sheets, which is making it a bit hard to follow. I also notice that none of the references of publications include a DOI. (as a separate point, will these publications receive a formal citation?)

- Tables in general: not all of the abbreviations and codes used in the various tables were explained, and I had difficulty understanding some of the tables. It did not help that the file names were randomised by the submission system (I assume), so it took me some time to identify which table was which. However, a cover sheet for each table or simply a table name and caption within the table files would have helped a lot.

- Figure in suppl section 3.3.1(?) 'Origination and scaled diversity' (page 13): unclear which figure panel shows what (left vs right-hand sides have identical labels as far as I can tell)

Reviewer #2 (Remarks on code availability):

I have verified the code repository on GitHub: all relevant files seem to be included and accessible, and well described in the

readme. The code itself looks tidy and suitable commented. However, as I am not a regular R user, I did not attempt to run the code on my local machine, so cannot guarantee that it reproduces the results.

Please note that the repository contains a copy of the supplementary files submitted with this manuscript (which should be updated during author revisions).

Reviewer #3 (Remarks to the Author):

I thoroughly enjoyed reading the manuscript “Fossils for Future: the billion-dollar case for palaeontology’s digital infrastructure”. I believe it is an excellent and timely contribution that tackles one of the most pressing challenges in palaeontology and allied Earth sciences: the need for sustainable, interoperable, and equitable data systems. Specially when considering current accelerated digital revolution. It combines historical perspective, meta-research, and forward-looking policy guidance to demonstrate both the scientific and economic value of long-term data stewardship. The result is a work that is not only analytically robust but also conceptually ambitious and deeply relevant to the future of data-intensive science.

One of the major strengths is its empirical foundation. The meta-analysis (118 databases) with supporting code and data openly shared, provides a rigorous and transparent basis for the arguments developed. Also, the glossary. Well done! This is complemented by a impactful economic assessment of data replacement costs, which compellingly quantifies the tangible stakes of sustaining digital infrastructure, although the point is made that loss to in cases where re-collection is not possible can not be quantified. The manuscript also lays out a very much needed strategic vision: transitioning to modular, third-generation databases are clearly articulated, actionable, and well aligned with the broader trajectory of data-driven research in the Earth sciences.

By incorporating the perspectives of database developers, curators, and users, the authors ensure that their proposals are grounded in practical realities and informed by those directly involved in data creation and stewardship. The result is a document that not only diagnoses systemic challenges but also speaks directly to the people best positioned to address them. The text is written in a clear and absorbing style, with a logical progression that moves flawlessly from historical context to future pathways. Figure 2, in particular, is very informative and helps the reader visualize the evolution of palaeontological data infrastructures over time. The manuscript’s commitment to open science...through the use of openly available data, code, and transparent methodologies... strengthens its contribution and sets a worthy example.

That said, a few points could be considered. These are suggestions only! When describing palaeontology’s interdisciplinary reach (“This scientific inquiry draws from geology, biology, chemistry, archaeology, and mathematics...”), it would be valuable to explicitly highlight the immensity and complexity of geological timescales. Emphasising how palaeontology invites us to transcend human temporal perspectives and contemplate Earth’s vast evolutionary history would underscore the transformative nature of the field. This provides an invaluable aspect of the fossil record, linked to curiosity, discovery and contemplation beyond more pragmatic aspects. Similarly, in the section on the evolving interpretation of the fossil record, the discussion could be expanded to include examples of how reinterpretations inform phylogenetic calibration or the validation of spatially explicit eco-evolutionary models, illustrating the iterative and integrative character of contemporary palaeoscience.

The manuscript’s reflections on governance are particularly strong, but the point could be made even more powerfully by noting that empowering those who create and curate palaeontological data not only addresses issues of access, provenance, and equity but also extends the longevity and resilience of datasets. In the same spirit, the authors might consider discussing how so-called “extinct” datasets could still be rescued or revived. Exploring strategies for salvaging abandoned or obsolete data—essentially “rescuing digital fossils” would add an inspiring and actionable dimension to the argument.

Table 2 is another great element of the paper. It moves beyond theory to provide concrete guidance on how to act, following a clear and logical progression from design principles to governance, and it connects directly to the main text. If the authors wish to strengthen it even further, they might consider adding elements such as capacity-building programs, long-term digital preservation and risk management strategies, mechanisms for regional and linguistic equity, structured policy engagement frameworks, and citizen science or outreach initiatives. These would reinforce the manuscript’s emphasis on sustainability and broaden the scope of its recommendations.

Language-wise, the manuscript is well written and accessible, but there are a few opportunities to simplify terms without losing precision. For example, “nimble” could be replaced with “quickly,” “efficiently,” or “flexibly,” “extant data” with “existing data,” “interlocking elements” with “connected components,” and “per capita extinction rate of databases” with “average database extinction rate.” Such adjustments would improve readability for non-native speakers and early-career researchers.

Finally, the recommendations in Tables 2 and 3 are strong, but their real-world impact could be enhanced by structuring them as short-, medium-, and long-term priorities or by grouping them according to stakeholder type (e.g., researchers, curators, funders, policymakers). A brief comparison of palaeontology’s digital infrastructure with other data-intensive fields such as genomics or astronomy would also enrich the discussion, potentially revealing lessons and strategies that could be adapted from those disciplines.

In conclusion, this is a methodologically rigorous, conceptually forward-looking, and highly significant manuscript. It makes a compelling and well-supported case for sustained investment in palaeontological digital infrastructure and provides a thoughtful roadmap for building systems that are not only scientifically powerful but also equitable, sustainable, and open. I strongly commend this manuscript for publication.

Reviewer #3 (Remarks on code availability):

very clean git. Useful and direct to the point.

Version 1:

Decision Letter:

7th November 2025

Dear Dr. Dowding,

Thank you for submitting your revised manuscript ""Fossils for Future: the billion-dollar case for palaeontology's digital infrastructure" (NATECOLEVOL-25093173A). It has now been seen again by the original reviewers and their comments are below. The reviewers find that the paper has improved in revision, and therefore we'll be happy in principle to publish it in Nature Ecology & Evolution, pending minor revisions to comply with our editorial and formatting guidelines.

If you have not done so already, please ensure that you also email us a completed copy of the Reporting summary :

Reporting summary: https://www.nature.com/documents/nr-reporting-summary.pdf

[redacted]

Reviewer #1 (Remarks to the Author):

I am satisfied with the minor revisions completed by the authors.

Reviewer #2 (Remarks to the Author):

Thank you for addressing my previous comments and those of the other reviewers. I am happy to be able to recommend the revised manuscript for publication "as is".

Marthe Klöcking
(Dr., not Prof.)

Reviewer #3 (Remarks to the Author):

All comments have been fully addressed with clear, thoughtful and well-integrated revisions. The response document shows nimble refinement across all sections, ensuring that every point raised has been resolved fully and coherently.

Version 2:

Decision Letter:

13th January 2026

Dear Dr Dowding,

We are pleased to inform you that your Analysis entitled "The billion dollar case for sustaining palaeontology's digital databases", has now been accepted for publication in Nature Ecology & Evolution.

Over the next few weeks, your paper will be copyedited to ensure that it conforms to Nature Ecology and Evolution style. Once your paper is typeset, you will receive an email with a link to choose the appropriate publishing options for your paper and our Author Services team will be in touch regarding any additional information that may be required

Due to the importance of these deadlines, we ask you please us know now whether you will be difficult to contact over the next month. If this is the case, we ask you provide us with the contact information (email, phone and fax) of someone who will be able to check the proofs on your behalf, and who will be available to address any last-minute problems . Once your paper has been

scheduled for online publication, the Nature press office will be in touch to confirm the details.

Acceptance of your manuscript is conditional on all authors' agreement with our publication policies (see www.nature.com/authors/policies/index.html). In particular your manuscript must not be published elsewhere and there must be no announcement of the work to any media outlet until the publication date (the day on which it is uploaded onto our web site).

Authors may need to take specific actions to achieve compliance with funder and institutional open access mandates. If your research is supported by a funder that requires immediate open access (e.g. according to [Plan S principles](https://www.springernature.com/gp/open-science/plan-s-compliance) or the [NIH public access policy](https://www.springernature.com/gp/open-science/us-federal-agency-compliance)) then you should select the gold OA route, and we will direct you to the compliant route where possible. Because authors warrant under our subscription licensing terms that they haven't committed to licensing any version of their article under a licence inconsistent with the terms of our agreement – including the applicable embargo period – publication under the subscription model isn't suitable for authors whose funders require no embargo.

We welcome the submission of potential cover material (including a short caption of around 40 words) related to your manuscript; suggestions should be sent to Nature Ecology & Evolution as electronic files (the image should be 300 dpi at 210 x 297 mm in either TIFF or JPEG format). Please note that such pictures should be selected more for their aesthetic appeal than for their scientific content, and that colour images work better than black and white or grayscale images. Please do not try to design a cover with the Nature Ecology & Evolution logo etc., and please do not submit composites of images related to your work. I am sure you will understand that we cannot make any promise as to whether any of your suggestions might be selected for the cover of the journal.

You can generate the link yourself when you receive your article DOI by entering it here: <http://authors.springernature.com/share>.

[redacted]

P.S. Click on the following link if you would like to recommend Nature Ecology & Evolution to your librarian <http://www.nature.com/subscriptions/recommend.html#forms>

** Visit the Springer Nature Editorial and Publishing website at http://editorial-jobs.springernature.com?utm_source=ejP_NEcoE_email&utm_medium=ejP_NEcoE_email&utm_campaign=ejp_NEcoE for more information about our career opportunities. If you have any questions please click [here](mailto:editorial.publishing.jobs@springernature.com).**

Response to Reviewers for NATECOLEVOL-25093173

Fossils for Future - Dowding et al

Dear Editor and Reviewers,

We thank you all for your insights and constructive comments

The response to reviewers is structured so that our responses directly follow points raised by the reviewers, formatted within text boxes. Where possible, text changes are included within the response textboxes. Structural changes associated with the introduction of the methods section and discussion of results and pointed towards, rather than quoted in response to reviewers.

Kind Regards,

Elizabeth Dowding on behalf of the authors.

Reviewer #1 (Remarks to the Author):

Although I don't work in paleontology, I have dealt with building and integrating large databases in other contexts, and I found this work to be an interesting pitch for advancing database infrastructure. In the sub disciplines I am familiar with, probably the exact same narrative could be made about generations 1-3 identified by the authors. The manuscript has a clear and concise narrative and seems quite polished. Just a few random thoughts came to mind:

Is it worth mentioning the need for redundancy in skills across the community? Modularity partly addressing this by reducing the diversity of infrastructures that database developers need to understand, and community organized databases make this more feasible by distributing the effort across multiple research groups, but is it worth addressing the need for redundancy explicitly?

We thank the reviewer and highlighted in our revision that modularity would address the problem of redundancy.
Line 356: "To address data fragmentation and structural redundancy in databasing effort"

I like the back of the envelope calculation for value; provides a nice teaser for the title and is easy to understand in the text.

I like the specific mention of the cyberattack, as it really isn't something we tend to account for in database development and could be increasingly important, particularly as the value of the database grows.

Thank you for these positive comments!

Are there any more attributes of databases that have gone extinct besides 5 year funding cycle lifespans? Such attributes might help identify other strategies to enhance 3rd generation efforts

Here and in association with the comments of Reviewer 2, we have included text to suggest that community buy-in (e.g. through data contributions, and international collaboration), and a generalist scope (e.g. intaking data from all fossils, such as the Paleobiology database) are likely important differences between the long-lived and short-lived databases.

From Line 369: Significant features of long-lived databases include international collaboration in data stewardship (e.g. Neotoma), significant community contributions by way of volunteered data, and a generalist scope for data ingestion (see Supplementary).

I wouldn't mind seeing a few more explicit examples of insights gained as a result of aggregating paleo data to make the value more tangible and address more specifically what we have gotten for \$3 billion. E.g., could you make the case that entire sub disciplines of paleontology have emerged, or at least been enabled, from connecting databases?

An excellent suggestion.

From line 227: Careers of an entire generation of scientists are now based on publicly-accessible, interoperable data, and the access to international, high-quality data has led to the rise of palaeobiology and conservation palaeontology as quantitative subdisciplines^{61,62}. Similar stories are also evident in allied fields, such as geochemistry, where PetDB and GEOROC enabled the rise of 'statistical geochemistry'⁶³.

It seems like there's some successes and guidance to be borrowed from genbank and GBIF - what might those be for paleo data?

Space limitations make the direct explanation difficult, instead GBIF in particular has been highlighted as a model to follow for future directions.

From Line 220: These scientific needs demand further advances in how palaeontological data are reported, structured, integrated, managed, and sustained. Cross-institutional aggregation of museum specimen information into iDigBio⁵⁴ and the Global Biodiversity Information

Facility (GBIF55) provide models of how biodiversity databases can grow and be enhanced by ever-improving biodiversity data standards, such as the Darwin Core56 and ABCDEFG57, featuring a stronger focus on available metadata58.

From Line 480 There is a new paragraph to highlight successes from other fields and data structures.

Reviewer #2 (Remarks to the Author):

Dowding et al. present a review of the data landscape for palaeontology, and derive recommendations to ensure continued relevance and future sustainability of these data systems. Given the rise in 'big data' and machine learning approaches across all sciences, data literacy and awareness of our global data infrastructure are becoming essential to all researchers. This review, therefore, is highly topical and very relevant to the audience of Nature Ecology & Evolution. However, I have a few concerns/suggestions, especially regarding the first half of the manuscript. I recommend publication of this article in Nature Ecology & Evolution following minor revisions.

We thank Prof. Klöcking for their review and suggestions.

General comments:

1. The introduction and first half of Section 2 seem overly simplified and general to me. Whilst the second half of the manuscript is very well written and has a clear story line, these first few pages lack that same focus: key topics in palaeontology and data science are introduced, but the connection is not always clear and I often found myself wanting a slightly more detailed explanation of specific concepts. I would recommend shortening the general part of the introduction to only the most relevant topics, and instead giving a few more details and examples on the concepts that are introduced and how they connect.

We have looked for ways to clarify the Introductory sections 1 & 2. However, we have not made drastic reductions in this section, given that we are trying to make this paper useful for a broad range of audiences, ranging from other database curators to non-specialists from allied disciplines and within funding organisations and governments. We also believe that the three-stage summary of database development history (Section 2) is a useful and novel contribution of this manuscript that may carry parallels in other disciplines (see Reviewer 1).

2. In contrast, Section 3, i.e. the new results presented in this article, seems a little too well summarised and a lot of the detail of the database meta-analysis is 'hidden' in the supplementary materials. I understand that the journal format does not allow for all methodological detail to be included in the main text. However, at present, this section came across as fairly cursory, back-of-the-envelope type calculations; and only after looking through all supplementary tables did the impressive scale of the work become apparent to me. If possible, I would recommend moving a few more details from the supplement into the main text here, perhaps at the expense of Sections 1 and 2.

The Article type has been moved from a Review to an Analysis. The methodological framework and interpretation of results has now been given more space within the article.

3. Your results on database extinction rates are very sobering. You already give multiple reasons of why most data systems are set up to fail in the short- to mid-term. Can you say more about the success strategies of what enabled the few long-lived data systems to persist? You hint at possible strategies in Section 2 and in your recommendations, but I was also hoping to see an analysis of what set the long-lived systems apart from the others (and what they have in common). This information is already contained within the supplementary 'CuratorReview.xlsx', I believe, but never explicitly stated. Another aspect you only hint at is international cooperation, governance and funding: all successful data systems in other disciplines have some level of internationality in common – I think this would also be useful to add to your discussion.

In association with the comments of Reviewer 1, we have included text to suggest that community buy-in (e.g. through data contributions, and international collaboration), and a generalist scope (e.g. intaking data from all fossils, such as the Paleobiology database) have been significant structural differences between the long-lived and short-lived databases.

A new paragraph has been added from 319: Databases achieving longevity exceeding 20 years tended to employ one or more of three distinct strategies....

Further, the new section on methods and results now follows these suggestions. Thank you!

More detailed comments are appended below that I hope will help to address these general points.

Kind regards,
Marthe Klöcking

Detailed Comments

Section 1, first paragraph, line 10: missing opening parenthesis in reference to the Glossary. This sentence (lines 10-15) is very long and a bit hard to follow.

Amended.

From line 67: From their very beginning, palaeontological databases (see Glossary, Supplemental Table 1) played pivotal roles in enabling the field to scale up from site-level studies to global-scale research. [...]

Second paragraph, line 1: I would suggest the word 'archives' rather than 'repositories' here, see definitions in the CODATA RDM Terminology: <https://vocabs.ardc.edu.au/viewById/685>.

Thank you! Amended throughout the text.

Third paragraph, lines 5-6: replace 'databases and systems' with 'data systems'? Please explain what is meant by 'diversity dynamics'

Amended: 'data systems' accepted.

Diversity Dynamics is explained in the new Methods section.

Section 2.1, last paragraph, last sentence: 'provide evidence OF provenance'? This sentence is a bit unclear and could benefit from more detail/nuance. I can see two separate issues here: 1) the risk of erroneous data records in secondary databases when only using processed data; 2) the question of attribution and importance of preserving a record of data sources and provenance.

Accepted.

From Line 137: Lastly, good provenancing can ensure against corollary risks such as 'data cannibalism' 25,26 when databases are used as data sources for other, secondary databases without proper attribution and dataset-level provenancing, which can violate the standard CC-BY licenses that accompany most open-access data resources.

Section 2.2.1, line 4: missing closing parenthesis after ‘Sepkoski, 1982’

Accepted.

Section 2.2.2, third paragraph: the list of example databases is a bit hard to follow, I would recommend separating examples with a semi-colon or similar for clarity.

Accepted.

Section 2.3, first paragraph, lines 4-7: sentence a bit hard to follow, and very abstract. Can you give an example of where modular design has facilitated interoperability?

Section amended and reads as follows, highlighting a specific database, key examples from literature, and a focus on cross-disciplinary data pooling:

Section 2.3: From line 198.

Second paragraph, lines 5-9: what exactly are you referring to with ‘inequities’? Data access, data coverage and bias in distribution, etc ...? Can you give an example?

Amended.

From line 232: At the same time, new concerns have arisen about whether these databases encode and perpetuate past and present inequities, such as parachute science⁶², and how best to reduce these inequities to truly fulfill the deeper mission of these databases to ensure democratized data access for all^{61–65}.

Section 3.2, second paragraph, lines 6-8: do these or other systems in your analysis give any insights on joint, international governance? In my opinion this is one of the major hurdles to database sustainability: our science is global, but the funding tends to be local or national (and competitive). I think you need to comment on whether the successful databases in palaeontology are big players supported by individual, currently well-off nations (that dominate the rest of the world) or if there are international cooperations. GBIF is a fantastic example, and there may be others?

Certainly! Amended. The call for a pluralist, international body seen in section 4.2, final paragraph also relates to this.

From line 282: is five-year timing coincides with the standard competitive funding program of many large research grants from wealthy international Unions or high Gross Domestic Product countries, e.g. through the European Research Council or the National Science Foundation of the United States of America, respectively.

*I also wonder if your discussion (either here or later) might benefit from more of a comparison with other disciplines. These are big questions that are being discussed by almost all data-driven research fields; we can definitely learn from each other, and your conclusions and recommendations are echoed in other disciplines. I am biased, of course, but Klöcking et al. (2025) give a similar review of the geochemical data landscape, including an estimate of the value of data systems, recommendations for the future and a comparison with success in other disciplines such as seismology and crystallography, which might be helpful as a starting point. Klöcking, M., Lehnert, K. A., & Wyborn, L. (2025). Geochemical databases. In *Treatise on Geochemistry* (pp. 97–135). Elsevier. <https://doi.org/10.1016/b978-0-323-99762-1.00123-6>*

I believe that could be a paper for future development, perhaps a collaboration across the Earth sciences built out of both our work? We have noted within the text that palaeontology is being used as an example of a concern felt by many disciplines.

That said, a paragraph has been added to the end of section 4.2 to highlight the success stories from other disciplines.

Figure 2: perhaps I missed it, but please explain (again) what you mean by 'database richness', especially in comparison to 'diversity' (i.e. number)

This has now been addressed in the methods section.

Section 4.2: I think somewhere in this section you need to comment on how the three databases that are older than 25 years have managed to survive so long. What are their business model(s) and why were they successful?

In addition to the changes made above, a further sentence has been added

From line 447: Long-lived databases have been developed and maintained through a combination of sporadic funding, international support, and unfunded volunteer/service work61,69,98. The persistence of these databases through all this financial precarity is a

testament to their importance and the work of many scientists to keep them going.

Third paragraph, line 6 onward: NFDI4Earth do not currently provide basic funding for data infrastructure (only for specific projects that advance interoperability), and its long-term existence is still uncertain. Perhaps better examples would be ARDC (<https://ardc.edu.au/>), AuScope (<https://www.auscope.org.au/>) and EarthScope (<https://www.earthscope.org/>); maybe also EPOS (<https://www.epos-eu.org/>)?

Thank you for the catch! Amended to the ARDC.

Line 9 onwards (and Table 2): You mostly discuss national funding here: how can we move to more international governance and sustainability?

The ultimate aim is certainly that, as suggested in the final paragraph of section 4.2. We hope the roadmap we suggest will be the building blocks for an international initiative, e.g. "Establish core infrastructure grants" and "Foster community governance". I believe these strategies are already being developed by the community, rather than imposed.

Section 4.3, third paragraph: not Figure 3 was missing from the manuscript files I had access to.

Amended!

Supplementary Materials:

- Table 1 (Glossary): are you aware of the CODATA RDM terminology (<https://codata.org/initiatives/data-science-and-stewardship/rdm-terminology-wg/>; <https://vocabs.ardc.edu.au/viewById/685>)? This glossary contains some widely used terminology, I wonder if it is worthwhile adopting existing definitions.

I have added a note in the supplementals that the glossary provides internally consistent terminology, rather than CODATA RDM terms. This decision was made in collaboration with the database maintainers within the authorship team, owing to disagreements about domain-specific knowledge. Consequently, usage is functional, not definitive.

- *Supp-Table 3 (Publication products): formatting seems inconsistent between the different table sheets, which is making it a bit hard to follow. I also notice that none of the references of publications include a DOI. (as a separate point, will these publications receive a formal citation?)*

Formatting has been addressed. Citation will be dependent upon discussion with the Editor.

- *Tables in general: not all of the abbreviations and codes used in the various tables were explained, and I had difficulty understanding some of the tables. It did not help that the file names were randomised by the submission system (I assume), so it took me some time to identify which table was which. However, a cover sheet for each table or simply a table name and caption within the table files would have helped a lot.*

- *Figure in suppl section 3.3.1(?) 'Origination and scaled diversity' (page 13): unclear which figure panel shows what (left vs right-hand sides have identical labels as far as I can tell)*

I have added labels/table names to address this.
'Origination and scaled diversity': label updated.

Reviewer #2 (Remarks on code availability):

I have verified the code repository on GitHub: all relevant files seem to be included and accessible, and well described in the readme. The code itself looks tidy and suitable commented. However, as I am not a regular R user, I did not attempt to run the code on my local machine, so cannot guarantee that it reproduces the results.

Please note that the repository contains a copy of the supplementary files submitted with this manuscript (which should be updated during author revisions).

Accepted. GIT and supplements both updated.

We thank you for these extremely valuable suggestions.

Reviewer #3 (Remarks to the Author):

I thoroughly enjoyed reading the manuscript “Fossils for Future: the billion-dollar case for palaeontology’s digital infrastructure”. I believe it is an excellent and timely contribution that tackles one of the most pressing challenges in palaeontology and allied Earth sciences: the need for sustainable, interoperable, and equitable data systems. Specially when considering current accelerated digital revolution. It combines historical perspective, meta-research, and forward-looking policy guidance to demonstrate both the scientific and economic value of long-term data stewardship. The result is a work that is not only analytically robust but also conceptually ambitious and deeply relevant to the future of data-intensive science.

One of the major strengths is its empirical foundation. The meta-analysis (118 databases) with supporting code and data openly shared, provides a rigorous and transparent basis for the arguments developed. Also, the glossary. Well done! This is complemented by a impactful economic assessment of data replacement costs, which compellingly quantifies the tangible stakes of sustaining digital infrastructure, although the point is made that loss to in cases where re-collection is not possible can not be quantified. The manuscript also lays out a very much needed strategic vision: transitioning to modular, third-generation databases are clearly articulated, actionable, and well aligned with the broader trajectory of data-driven research in the Earth sciences.

By incorporating the perspectives of database developers, curators, and users, the authors ensure that their proposals are grounded in practical realities and informed by those directly involved in data creation and stewardship. The result is a document that not only diagnoses systemic challenges but also speaks directly to the people best positioned to address them. The text is written in a clear and absorbing style, with a logical progression that moves flawlessly from historical context to future pathways. Figure 2, in particular, is very informative and helps the reader visualize the evolution of palaeontological data infrastructures over time. The manuscript’s commitment to open science...through the use of openly available data, code, and transparent methodologies... strengthens its contribution and sets a worthy example.

Our sincere thanks for your positive remarks and support!
--

That said, a few points could be considered. These are suggestions only! When describing palaeontology’s interdisciplinary reach (“This scientific inquiry draws from geology, biology, chemistry, archaeology, and mathematics...”), it would be valuable to explicitly highlight the immensity and complexity of geological timescales. Emphasising how palaeontology invites us to transcend human temporal perspectives and contemplate Earth’s vast evolutionary history would underscore the transformative nature of the field. This provides an invaluable aspect of the fossil record, linked to curiosity, discovery and contemplation beyond more pragmatic aspects.

We agree and have made small changes throughout to highlight the ‘softer’ and ‘higher’ benefits for palaeontological research, but the earliest is as follows:

Paragraph 1: This scientific inquiry draws from geology, biology, chemistry, archaeology, and mathematics, amongst others, to breach human temporal perspectives, reconstruct ancient ecosystems, investigate the drivers of biodiversity, and forecast how life will respond to today’s changing environments (Dietl & Flessa, 2011; Dillon et al., 2022; Kiessling et al., 2023).

Similarly, in the section on the evolving interpretation of the fossil record, the discussion could be expanded to include examples of how reinterpretations inform phylogenetic calibration or the validation of spatially explicit eco-evolutionary models, illustrating the iterative and integrative character of contemporary palaeoscience.

We tried to keep examples and language very general. One of the aims of the paper is function: to present it to funding bodies who are often made up of non-specialists. We selected examples that highlighted utility outside of standard palaeontological subdisciplines to strengthen our argument that palaeontological data is necessary across Earth Sciences.

The manuscript’s reflections on governance are particularly strong, but the point could be made even more powerfully by noting that empowering those who create and curate palaeontological data not only addresses issues of access, provenance, and equity but also extends the longevity and resilience of datasets. In the same spirit, the authors might consider discussing how so-called “extinct” datasets could still be rescued or revived. Exploring strategies for salvaging abandoned or obsolete data—essentially “rescuing digital fossils” would add an inspiring and actionable dimension to the argument.

An excellent point. Text has been amended, in addition to changes within methods and results interpretation.

The paragraph of section 4.1, beginning at line 376, has particularly been rewritten to address this.

Table 2 is another great element of the paper. It moves beyond theory to provide concrete guidance on how to act, following a clear and logical progression from design principles to governance, and it connects directly to the main text. If the authors wish to strengthen it even further, they might consider adding elements such as capacity-building programs, long-term digital preservation and risk management strategies, mechanisms for regional and linguistic equity, structured policy engagement frameworks, and citizen science or outreach initiatives.

These would reinforce the manuscript's emphasis on sustainability and broaden the scope of its recommendations.

Thank you for these suggestions, they are extremely helpful. The table has been revised, particularly in the description, to incorporate these suggestions. Further edits related to your comments below on both Table 2 and 3 are expanded upon there.

Language-wise, the manuscript is well written and accessible, but there are a few opportunities to simplify terms without losing precision. For example, "nimble" could be replaced with "quickly," "efficiently," or "flexibly," "extant data" with "existing data," "interlocking elements" with "connected components," and "per capita extinction rate of databases" with "average database extinction rate." Such adjustments would improve readability for non-native speakers and early-career researchers.

The text has been reworked to maintain precision and accessibility and these terms considered and where possible altered, to the exclusion of 'nimble', which we have elected to retain for personal style.

Finally, the recommendations in Tables 2 and 3 are strong, but their real-world impact could be enhanced by structuring them as short-, medium-, and long-term priorities or by grouping them according to stakeholder type (e.g., researchers, curators, funders, policymakers). A brief comparison of palaeontology's digital infrastructure with other data-intensive fields such as genomics or astronomy would also enrich the discussion, potentially revealing lessons and strategies that could be adapted from those disciplines.

New columns have been added to Table 3 to reflect these suggestions, particularly stakeholders. Priority term was elected against, as ideally these should all be developed as soon as possible, and have long-term impact.

A paragraph has been added to the end of section 4.2 to highlight the success stories from other disciplines.

In conclusion, this is a methodologically rigorous, conceptually forward-looking, and highly significant manuscript. It makes a compelling and well-supported case for sustained investment in palaeontological digital infrastructure and provides a thoughtful roadmap for building systems that are not only scientifically powerful but also equitable, sustainable, and open. I strongly commend this manuscript for publication.

Reviewer #3 (Remarks on code availability):

very clean git. Useful and direct to the point

Our sincere thanks.